# Aerosol Spray Deposition of Liquid Metal and Elastomer Coatings for Rapid Processing of Stretchable Electronics

**DOI:** 10.3390/mi12020146

**Published:** 2021-02-01

**Authors:** Taylor V. Neumann, Berra Kara, Yasaman Sargolzaeiaval, Sooik Im, Jinwoo Ma, Jiayi Yang, Mehmet C. Ozturk, Michael D. Dickey

**Affiliations:** 1Department of Chemical and Biomolecular Engineering, North Carolina State University, Raleigh, NC 27695, USA; tvneuman@ncsu.edu (T.V.N.); sim2@ncsu.edu (S.I.); jma23@ncsu.edu (J.M.); jyang46@xust.edu.cn (J.Y.); 2Department of Electrical and Computer Engineering, North Carolina State University, Raleigh, NC 27695, USA; bnkara@ncsu.edu (B.K.); ysargol@ncsu.edu (Y.S.); mco@ncsu.edu (M.C.O.)

**Keywords:** liquid metal, microparticles, stretchable electronics, aerosol spray deposition

## Abstract

We report a spray deposition technique for patterning liquid metal alloys to form stretchable conductors, which can then be encapsulated in silicone elastomers via the same spraying procedure. While spraying has been used previously to deposit many materials, including liquid metals, this work focuses on quantifying the spraying process and combining it with silicones. Spraying generates liquid metal microparticles (~5 μm diameter) that pass through openings in a stencil to produce traces with high resolution (~300 µm resolution using stencils from a craft cutter) on a substrate. The spraying produces sufficient kinetic energy (~14 m/s) to distort the particles on impact, which allows them to merge together. This merging process depends on both particle size and velocity. Particles of similar size do not merge when cast as a film. Likewise, smaller particles (<1 µm) moving at the same speed do not rupture on impact either, though calculations suggest that such particles could rupture at higher velocities. The liquid metal features can be encased by spraying uncured silicone elastomer from a volatile solvent to form a conformal coating that does not disrupt the liquid metal features during spraying. Alternating layers of liquid metal and elastomer may be patterned sequentially to build multilayer devices, such as soft and stretchable sensors.

## 1. Introduction

Flexible, soft, and stretchable electronics find applications in sensors, actuators, energy harvesters, and antennas. Liquid metals (in particular, gallium alloys with melting points near or below room-temperature) are attractive for such applications because they are both highly conductive and stretchable due to their fluidic nature [1,2,3,4]. Metals that are liquid can be patterned in unique ways that are not possible with solid metals, including room temperature deposition [5]. Gallium alloys are of particular interest due to the presence of a thin (2–5 nm) oxide shell that forms spontaneously in air [6,7]. This oxide layer provides mechanical stability that enables patterning in spite of the high surface tension of liquid metals.

There are a variety of methods available to pattern liquid metal. Since these approaches have been reviewed elsewhere [5], we focus here specifically on additive approaches to give relevant context to this paper. One common additive method is filling microfluidic channels. After fabricating a microchannel (by soft lithography, for example), liquid metal is injected into the channel [8,9,10]. Alternatively, the microchannel could be filled by placing the channel under vacuum [11,12]. Both methods can produce high resolution features (it is straightforward to produce features of 10–100s of microns). Yet, the fabrication of microchannels can be time consuming. In addition, the conductive patterns must follow a continuous fluid path, which makes patterning of smaller, discrete structures challenging.

Patterns may also be formed via selective wetting, in which a prepatterned substrate is locally wetted with liquid metal. This may be done by patterning a substrate with very thin layers of solid metals (such as copper [13] or gold [14]) which are then directly wetted by the liquid metal. The liquid metal will only wet to the patterned regions, forming high resolution patterns. Alternatively, a single substrate could be patterned to form non-wetting and wetting regions [15,16,17,18]. Likewise, topographic stamps can be used to pattern liquid metals in an additive fashion [19,20,21].

3D printing and direct-write printing of liquid metals provides the ability to quickly modify patterns and iterate on designs [22,23,24,25,26,27,28,29]. Despite its appeal, printing presents some challenges due to the rheological behavior and high surface tension of liquid metal. Consequently, it cannot be directly extruded into filaments without the use of rheological modifiers [30] or electrochemically altering the surface oxide [31]. Instead, printing liquid metals involves using shear or tensile forces to pull a meniscus of metal from a nozzle. Thus, printing requires very close contact between the liquid metal meniscus and the substrate [28,32]. This is easy with flat surfaces, but surfaces with topography require complex control schemes to maintain a constant gap between the nozzle and substrate [33]. This type of printing requires good adhesion between the metal and the substrate, yet studies show liquid metals do not adhere well to rough surfaces [17,34]. This issue has been partly addressed by applying electric fields between the metal and substrate to promote adhesion [35,36]. Another approach is to disperse liquid metal particles in a polymer matrix (e.g., silicone) to make a printable ink [37,38] though these materials requires some post-processing such as mechanical force or freezing to “sinter” the liquid metal particles and form a conductive path [39,40].

Liquid metals have been deposited by physical vapor deposition [41]. This approach is relatively slow due to the need for vacuum processing. This approach is also somewhat wasteful since the metal deposits on the walls of the chamber, which must be covered to prevent adverse interactions between the liquid metal and the chamber walls. Nevertheless, this approach shows promise for depositing liquid metal in a non-contact manner through the vapor phase onto a receiving substrate.

Likewise, drop-on-demand patterning, where droplets of liquid metal are placed or ejected onto the surface in a designed pattern, presents an approach which does not require direct contact with a substrate, and still allows designs to be rapidly modified and iterated upon. Preparing liquid metal micro- or nanoparticles is straightforward [42,43,44]. Ink jet printing of liquid metal particles has been demonstrated previously by jetting liquid metal nanoparticles suspended in a volatile solvent from a nozzle [45]. However, the patterns based on nanoparticles are not electrically conductive and require some post-processing steps to sinter the particles together. Sintering can be accomplished by mechanical force [45,46,47] or by exposure to a laser [48,49,50].

Alternatively, by aerosolizing liquid metal microparticles, a conductive film can be easily produced on a substrate. Spraying aerosolized liquid metal over a stencil produces conductive patterns rapidly and across a large area. Since not all the metal passes through the stencil, there is some waste, yet most of the metal blocked by the stencil can be recovered. This technique has been used successfully to fabricate soft electrodes [51,52,53], sensors [54,55], energy harvesters [56,57,58] and antennas [59] Motivated by these applications, this study examines how it forms robust electrical contacts, especially in comparison to inkjet processing. We also use the spray coating process to pattern thin encapsulating layers of silicone, which are important to ensure the liquid metal does not smear during handling after deposition. Silicone elastomers are used commonly to encase surface features by drop-casting the silicone or by spin-coating the silicone to produce very thin and uniform layers. Dilution of silicones within a volatile solvent (hexane, toluene, etc.) has been used to produce a solution which can be aerosolized and used to coat surfaces in complete and conformal encapsulation layers [54,60,61] This may be done to encase fluidic materials, prevent short circuiting between conductive elements, or to alter surface properties. In this paper we examine the characteristics of spray processing which lead to high conductivity patterns, as well as demonstrate simple, multilayer structures that can be fabricated exclusively via additive spray processing, which is promising for future roll-to-roll processing of flexible devices. Spray coating allows the silicone layers to be very thin while conforming to the shape of the existing surface features.

## 2. Results and Discussion

### 2.1. Processing Overview

In this work, we spray coat materials using an airbrush tool, commonly used for spraying dilute paints. The barrel of the airbrush holds a reservoir of liquid (e.g., liquid metal) and connects to a compressed air source. When triggered, the flow of air breaks the liquid metal into small particles that flow out of the nozzle with air. The nozzle directs a stream of particles onto a substrate until the particles cover the surface with a complete film. This process is compatible with a variety of substrates, as seen in prior work [52,54,55,56,59,62,63,64,65,66]. While others have shown that patterning is possible using a directed laser to ablate liquid metal from the surface [53], here we make use of vinyl stencils (also commonly used in the crafting community) adhered to the substrate to pattern the metal. We use a craft cutter (Cameo) that can rapidly fabricate complex openings in a vinyl sheet for stenciling (a technique called xurography [67]). Removing the stencil after spraying leaves behind the desired liquid metal pattern on the substrate. This process is depicted schematically in Figure 1a.

In comparison to serial patterning methods (direct-write printing or microfluidic injection), aerosol jetting is high throughput. Large areas (hundreds of square centimeters) can be covered in seconds using a single spray nozzle. Though the process may produce some overspray, the residual liquid metal is recyclable by exposing it to a strong acid or base. Such extreme pH solutions remove the oxide layer from the liquid metal, which allows it to bead up into a puddle. Thus, most excess metal can be quickly reclaimed and used again later.

The resolution of the resulting patterns is primarily limited by the resolution of the stencil (i.e., a higher resolution stencil will result in a higher resolution pattern). Using low-cost, disposable vinyl as a stencil material, we were able to produce features down to 300 µm with ease. We observed that wider patterns have a smaller relative standard deviation in linewidth (see Appendix A), although all the standard deviations fall between 19–35 μm for features ranging from 300–2100 μm. This standard deviation results from line edge roughness, which quantifies a limitation of this approach. Figure 1b shows a square spiral pattern with line widths of ~300 µm. Features below this threshold necessitate either a smaller stencil or an alternative patterning method.

The feature height may be controlled by increasing the number of coatings applied during processing. Figure 1c depicts the cross-sectional profile of sprayed EGaIn features with varying numbers of coatings. As expected, the height increased linearly with the number of applied coatings. A single layer was defined by spraying metal on the surface for approximately 2 s at a standoff distance of approximately 10 cm above the substrate. We found each layer to have an average maximum thickness of 11.2 ± 2.3 μm. The given parameters ensured complete coverage of the substrate for each layer of coating.

### 2.2. Impact of the Surface Oxide

We observed that the sprayed patterns exhibited high conductivity without any post processing. Figure 2a shows the result of spraying liquid metal to connect a power supply to an LED. In the first frame the circuit is incomplete. The sprayed liquid metal connects the two electrodes and completes the circuit, illuminating the LED.

We tested how the conductivity of the sprayed EGaIn compared to bulk EGaIn, and if the excess oxidation caused by aerosolization may result in a loss of performance. To do so, we deposited lines of varying width and measured the resistance across two electrodes at increasing distances apart. Figure 2b depicts the resistance as a function of the distance between electrodes. We observed a linear increase in the measured resistance with increasing length, as expected. In addition, wider traces exhibited a lower resistance than thinner traces. We measured the average trace height to be 25 μm via white light interferometry, allowing us to calculate the cross-sectional area of each trace. We used the measured resistance values to calculate the resistivity of the patterns via Pouillet’s law, given in Equation (1).
(1)$$R=\frac{\rho l}{A}$$
where *R* is the resistance in Ohms, *ρ* is the resistivity, *l* is the distance between electrodes, and A is the cross-sectional area of the trace. The *A* term is the product of the width and height of the trace. Given that the trace height is over an order of magnitude less than the width, we assume a rectangular cross-section profile.

Fitting of the data in Figure 2b provides an estimate of the material resistivity. We calculated a value of 32.5 ± 6.3 × 10^−6^ Ω∙cm for the resistivity of sprayed EGaIn, which is within 10% of the literature reported value of 29.4 × 10^−6^ Ω∙cm [68]. These measurements are summarized in Appendix A. This indicates the presence of excess oxide has little impact on the resulting conductivity of the liquid metal patterns. By plotting the resistance as a function of trace length divided by cross-sectional area, the curves in Figure 2b collapse into a single linear plot, shown in Figure 2c. The slope of this plot corresponds to the resistivity of the EGaIn traces.

Figure 2d shows a comparison of three methods to deposit liquid metal through a stencil: spraying liquid metal on an untreated glass slide, spraying onto a glass slide in an HCl rich environment, and drawing a volume of liquid metal down across a slide. The spray coating creates a nice film across the untreated slide and the metal does not dewet during handling. To observe what happens in the absence of native oxide on the liquid metal, we pretreated the glass slide with 1 M HCl. Acidic and basic conditions remove the oxide on the surface of liquid metal, which exposes bare metal with high interfacial tension [69,70,71,72,73,74]. Spraying onto a glass slide in an HCl rich environment showed poor adhesion, and the liquid metal did not coalesce into a film. Rather, it remained as distinct droplets disconnected from each other. These differences, which are expected, are apparent in the optical micrographs in Figure 2d.

We also compared stenciling the same metal via a draw bar, in which a volume of bulk liquid metal was placed at one end of the stencil and drawn across the surface to cover the stencil. After removing the stencil, the liquid metal did not adhere fully to the surface, leaving large regions disconnected and non-uniformly wetted with metal. This finding suggests that the tension of the interface of the oxide-coated metal did not readily make contact through the stencil to the exposed regions of the substrate.

Taken in sum, we conclude that spraying produces better features than draw-down of metal across a stencil. The droplets ensure the metal passes through the stencil. When they impact the surface, the oxide ruptures and then reforms, pinning the drops to the surface and preventing dewetting. This is very important for fabricating stretchable circuitry which must be robust enough to withstand multiple strain cycles without failing [56,75].

This result also shows that aerosol spray coating of liquid metals produces patterns of high conductivity, which are not negatively impacted by the presence of excess oxide, and in fact rely on the oxide to adhere to the surface during deposition. Next, we consider why this process readily forms conductive networks, whereas nanoparticle inks require some post-processing sintering to form conductive paths. We hypothesize that this difference is reliant on two factors: particle size and speed.

### 2.3. Particle Rupture

#### 2.3.1. Effect of Particle Size

Prior work on gallium nanoparticles (tens to hundreds of nanometers) has shown that nanoparticle films require force to form a conductive network, and that smaller particles are stiffer than larger particles due to the mechanics of the native oxide skin [6,45,76,77,78]. Meanwhile, work which features metal droplets that are tens to hundreds of microns in diameter are capable of immediately and repeatably forming conductive paths [71,72,79]. Thus, we sought first to confirm the size of the particles produced via the spray deposition process. The particles were collected by spraying liquid metal into a bath of ethanol to prevent particles from rupturing on impact. After collection, the ethanol was evaporated, and the remaining particles were observed via optical microscopy and sized using ImageJ (see Appendix A). The droplet particles had an average diameter of 5.8 ± 3.2 µm, shown in Figure 3a and the particle size follows a log-normal distribution, which is expected for aerosols [80].

To systematically assess the behavior of liquid metal micro- and nano-particles, we prepared suspensions of particles in solution and cast them as films. Nanoparticles were prepared via sonication of liquid metal in ethanol, following the procedure set forth by previous work in our group on liquid metal nanoparticles [81,82]. Microparticles were prepared by dispersing liquid metal in ethanol using a Vortex mixer.

We observed that all the films were natively insulating. Figure 3b depicts particles separated by the oxide layer. At a critical applied force, the particles sinter to form a conductive path. A mechanical probe with two copper electrodes was placed in contact with opposing ends of the films. The probe was connected to an Instron mechanical testing system to measure the force while a Keithley electrometer measured the current passing through the particles. All of the prepared films were natively insulating, and only passed a current after a critical load was applied, as seen in the plot in Figure 3c. A constant voltage is applied to the probes, but no current is measured initially until the particles rupture and merge, causing the massive spike in current. As anticipated, the largest particles (diameter of 191.3 ± 14.4 μm) required the least force to merge, showing a measurable current at a load of 0.98 N. The second group of particles (14.9 ± 1.3 μm) became conductive at a load of 13.5 N, and smaller particles (2.1 ± 0.3 μm) became conductive at a load of 98.5 N. The smallest set of particles (1.4 ± 0.1 μm) became conductive at a load of 125.3 N. The necessary load to induce conductivity is summarized in Appendix A.

Figure 3d is an optical micrograph showing the difference between a compressed (and therefore conductive) region and the initial uncompressed (insulating) state of the particles. The compressed region is clearly metallic and highly reflective, while the uncompressed region is still primarily a collection of individual particles, approximately 1 μm in average diameter. This experiment confirms that particles of larger size require less force to rupture than smaller particles, yet size alone cannot account for the behavior seen with spray coating liquid metal. Since sintering only occurs with applied force, we next sought to confirm that the force of impact of a sprayed droplet with a surface would provide sufficient force to rupture the oxide.

#### 2.3.2. Effect of Particle Speed

After observing that liquid metal microparticles, such as those seen in the aerosol spray process, will not spontaneously form a conductive pathway without some outside force, we expect to observe particles merge when impacting the surface with sufficient speed, as depicted in Figure 4a. A particle with sufficient kinetic energy at impact should rupture the oxide shell surrounding the particle (a velocity we call “rupture velocity”) and allow it to deform and merge with other particles to form a conductive film.

We consider both the particle falling at its terminal velocity, as well as the experimental speed of the aerosol particles. We calculated the experimental particle speed using a high-speed camera by timing the duration for particles to travel a known distance by observing particles impacting at two surfaces (glass slides). The experimental setup for measuring the speed is depicted in Appendix A, which measured a speed of 13.8 m/s. Since the fastest traveling particles are the ones to first reach the substrate, this represents an upper bound.

##### Rupture Velocity

We consider here two methods for the oxide to rupture. First, we examine the observation by Boley et al. that compressing a film of liquid metal particles to ~50% of their initial height produces a conductive network [45]. The energy necessary to cause this deformation (depicted in Appendix A) is related to the effective interfacial tension of the droplet, as described by Equation (2),
Δ*E_surface_* = *γ*Δ*A*(2)
where Δ*E_surface_* represents the change in surface energy of the droplet, *γ* is the effect surface tension, and Δ*A* is the change in surface area due to droplet deformation. By assuming the droplet distorts on impact from a spherical droplet to an ovoid with constant volume, we can calculate the change in surface area to be a function of the radii. Thus, we can calculate the required energy to compress a droplet to 50% the initial height. Neglecting the impact of viscous dissipation through the fluid, the energy for deformation will be provided exclusively from the kinetic energy of the droplet prior to impact.
(3)$$\Delta E_{K}=\frac{1}{2}mv^{2}=\frac{2}{3}\pi \left(\rho r_{0}^{3}\right)v^{2}$$

By equating the kinetic energy of a droplet (Equation (3)) to the required change in surface energy (Equation (2)), we arrive at an equation which describes the velocity required for a droplet to sufficiently deform at impact.
(4)$$v_{rupture}=\;\sqrt{\frac{1.38\gamma }{\rho r}}$$

Figure 4b shows this equation plotted as a function of particle diameter and compares it with the predicted terminal velocity of a falling particle. At the experimental particle speed of 13.8 m/s, we predict that particles greater than ~2 µm in diameter will be sufficiently deformed on impact to rupture and form a conductive network. The region of Figure 4b highlighted in red shows theoretical conditions to produce particles that have sufficient energy to break the oxide on impact.

##### Surface Stress at Impact

Alternatively, we consider the velocity needed to exceed the critical stress needed to rupture the surface layer of gallium oxide on each particle upon impact. Literature reports that the oxide breaks beyond a critical surface stress of ~0.2–0.6 N/m [8,83,84]. Assuming that the kinetic energy of the drop at impact transfers exclusively to transfer energy to the droplet surface (i.e., neglecting any viscous dissipation, energy transfer to the substrate, or changes in potential energy), it is possible to predict what minimum kinetic energy a droplet must have to surpass the critical surface stress. The surface stress can be estimated by dividing the kinetic energy (*E*) of the particle by the surface area (*A*).
(5)$$\sigma_{surface}=\frac{KE}{SA}=\frac{\frac{1}{2}mv^{2}}{4\pi r^{2}}=\frac{\rho rv^{2}}{6\pi }$$

Any particles that experience a surface stress beyond the critical value of ~0.5 N/m should rupture and merge together, forming a conductive network. Figure 4c depicts the surface stress arising from particles impacting a surface plotted as a function of particle diameter. The impact stress for a particle moving at terminal velocity is significantly less than a particle of the same size moving at the observed speed of ~14 m/s due to the lower momentum of the particle. The red highlighted region shows the conditions which possess sufficient kinetic energy at impact to rupture the oxide, which is important for forming a conductive network. This analysis predicts that at 14 m/s, particles greater than ~4 μm in diameter will rupture upon impact, in agreement with our previous analysis. Since the particles produced during aerosolization of liquid metal are on average 5.8 μm, this falls in the expected range to form a conductive path with no required post-processing (i.e., sintering).

We further confirmed this analysis by spraying liquid metal particles of smaller size via the same method. Because the airbrush is not capable of producing such small particles, the particles were prepared via sonication in ethanol and the suspension was loaded into the airbrush and deposited. Spraying of 1.4 ± 0.1 μm diameter particles on a glass slide produced a non-conductive film. The particles did not form a conductive network after spraying, supporting the hypothesis that both size and speed are required, and that very small (<1 μm) particles moving at high speed still do not form a conductive path.

### 2.4. Encapsulation via Spray Coating

For many applications, it is desirable or necessary for liquid metal structures to be encased in an insulating layer to prevent damage to the pattern, short circuits, or leakage of liquid metal onto other components during handling. This encasement can be done, for example, by drop-casting of a liquid elastomer pre-cursor over the liquid metal patterns followed by curing [22,82,85], or alternatively by spray coating [54,60,61]. Processing both conductive liquid metal patterns and insulating, elastomeric layers by spray coating provides a relatively straightforward path toward roll-to-roll processing of stretchable electronics in the future.

As an encasing layer, we used commercially available polydimethylsiloxane (PDMS, Sylgard-184 or Sylgard-186, Dow), which are thermally curable silicone elastomers commonly used in microfluidics and soft devices [86]. Dilution of the silicone liquid pre-polymer with a volatile solvent (e.g., toluene) results in a low viscosity fluid suitable for spray coating. During spraying, the volatile component evaporates, thereby coating a thin layer of elastomer on the substrate. The layers produced by this method can be both thin and conformal, conforming to any pre-existing features on the surface as shown in Figure 5a. In comparison, pure silicone—which is too viscous for spray coating of thin films—can produce thin layers via spin-coating or doctor blading, but such methods create a planar layer over preexisting surface features (Figure 5a).

Figure 5b shows the thickness of sprayed polymer films as a function of the number of coatings. A coating in this case is defined as a single manual pass across a glass slide (75 mm long and 50 mm wide), lasting approximately 5 s, and sprayed from 10 cm away at a pressure of 30 psi. As expected, additional coatings create thicker films. Interestingly, the lowest concentrations of polymer (20% Sylgard 186 or 40% Sylgard 184) produced the thickest films. This trend is opposite of the trend observed when spin-coating polymers, in which higher concentrations produce thicker films (see Appendix A). The lower viscosity of the more dilute samples flows more readily through the nozzle at a given pressure, resulting in a thicker layer of deposited silicone. Less dilute compositions may exhibit clogging in the nozzle and require greater pressure to push the fluid out of the nozzle. The result is less material flow overall, and thinner layers. A semi-log plot of the average layer thickness as a function of the prepared solution viscosity (Figure 5c) shows a linear trend of decreasing layer thickness with increased viscosity. Additionally, we observed that films prepared using higher viscosity starting material formed films with a higher degree of surface roughness (see Appendix A).

Figure 5d shows an array of liquid metal cones printed on a glass slide and then encased via spray coating of PDMS. The cones maintain their shape even after being contacted by a finger, as seen in the second image. As the goal of the elastomer layer was to produce thin layers that would contain the liquid metal structures, we sought to measure the force these layers could withstand before rupturing. This force was measured by pressing a probe (connected to a load cell) against a membrane of silicone while recording force displacement data. The probe is 13 mm in diameter with a semispherical tip. A clamping apparatus held the polymer membranes taut across a 30 mm circular opening. The experimental apparatus is shown in Figure 5e. The probe was driven into contact with the film at a rate of 3 mm/min and then continued to push against the film until the film ruptured or the probe reached the extension limit (90 mm).

We tested films made of both Sylgard 184 and Sylgard 186, diluted to 40 or 60 wt % before spraying. The films had an average thickness of 257.3 ± 31.1 μm. Figure 5f plots the maximum load and extension for each film at the point of rupture. The films handled a comparable load of 15 to 20 N before rupturing. However, the films made of Sylgard 186 extended to more than twice the length than Sylgard 184 before rupturing. This is expected from the higher reported toughness of Sylgard 186 [86].

### 2.5. Function as Soft Sensors

#### 2.5.1. Strain Sensor

We demonstrate here a set of resistive strain sensors fabricated via spray coating. Using vinyl stencils makes design iteration simple and implementing additional complexity in designs comes with no additional complexity in processing. Figure 6a shows an example of a device prepared via spray coating and undergoing strain. The device was prepared by placing a stencil over a silicone substrate (Dragonskin 10 Slow by Smooth-On). Both the silicone and liquid metal are soft and highly stretchable materials, well suited to use as strain sensors.

Figure 6b plots the resistance during strain cycling over time of the strain sensor shown in Figure 6a. The sensor provides consistent changes in resistance over many strain cycles. At 50% strain, the sensor showed an increase of ~86% in resistance. At 75% strain, the resistance increased by ~168% from the unstrained state.

We also demonstrate the simplicity with which alternative designs can be fabricated. Figure 6c shows four different sensor designs. The changes in patterning result in different responses to strain, as shown in Figure 6d. We cycled all four samples from 0 to 50% strain while measuring the resistance. Sensor 3 exhibited the greatest change in resistance (as expected given the path length parallel to the strain direction), showing a 35% increase in resistance. In comparison, Sensor 2 only showed ~5% increase under the same strain conditions since most of the length of the conductor aligns perpendicular to the strain direction. Sensors 1 and 4 exhibited similar behavior, showing a 22% and 30% increase in resistance under strain, respectively. The diversity of responses highlights the versatility of spray coating as a patterning technique and emphasize that the additional complexity of the pattern does not require any additional steps in sample preparation.

#### 2.5.2. Multilayer Capacitive Sensor

We also fabricated multilayer devices featuring alternating layers of PDMS and EGaIn, all deposited by spray coating. A multilayer capacitive sensor is shown in Figure 7a. This sample was prepared on a glass slide with a thin layer of PVA spin-coated on the surface to allow for easy detachment of the final device. A base layer of PDMS was deposited over the substrate via spray coating. After curing at 100 °C for 6 h, EGaIn was deposited on the substrate, which was then encapsulated with a second spray coated PDMS layer. This process was repeated to produce fully encased, multilayer, liquid metal electrodes.

We then characterized the response of the sensor by measuring the change in capacitance when contacted. The capacitance is defined by the following equation
(6)$$C=\frac{{Ɛ}_{0}{Ɛ}_{r}A}{d}$$
where *C* is the capacitance, *Ɛ*_0_ is the permittivity of free space, *Ɛ_r_* is the dielectric constant of the insulating material (PDMS), A is the area of the electrodes, and *d* is the distance between them. Since *Ɛ*_0_ and *A* remain constant, we see that capacitance is inversely related to the distance between electrodes (*C* ∝ 1/*d*). Applying compressive stress to the sensor pushes the electrodes closer together, thereby increasing the measured capacitance. Figure 7b shows the region where the two EGaIn electrodes overlap. The overhead view shows that the features do not merge and remain separated due to the conformal nature of the spraying process. The cross-sectional view shows the thin dielectric layer of PDMS separates the EGaIn layers.

Figure 7c plots the normalized change in capacitance as a function of the force applied. The overlapping area of this sensor is 25 mm^2^ (each side measuring 5 mm). The capacitance of the sensor without any contact was measured to be 25.3 pF, and the plotted values represent the deviation from this initial value. A linear fit of the data has a slope of 0.12 N^−1^. The probe used for this experiment is the same shown in Figure 5e and the applied force was measured using the Instron mechanical testing system in compression mode. The sensor may be made more sensitive by increasing the area of the electrodes, using a softer elastomer, or by using a material with a higher dielectric constant [87,88,89]. Future works may consider further characterization of sensors such as the one reported here. Variation of the PDMS dielectric layer thickness and material may provide different sensitivities to force and may be implemented in a variety of applications.

## 3. Conclusions

This work characterizes and demonstrates aerosol spray deposition of both liquid metal and silicone to form robust electrical contacts and patterns. The resulting patterns exhibit near bulk electrical conductivity, despite excess oxidation that may occur as a result of aerosolizing the liquid metal into microparticles. We also showed that this process relies on the presence of the oxide to adhere to the target substrate.

We demonstrated the effect of particle size and velocity on the processing of liquid metal via spray deposition. Particles must possess enough kinetic energy to rupture upon impact with the surface. Larger particles require less force to rupture and form a conductive pathway, but even films of particles an order of magnitude greater than those produced by spraying are insulating until a force of ~1 N is applied across the electrode area (1 cm by 5 cm). At the experimental particle velocity of 13.8 m/s, particles below 2–5 μm in diameter do not possess sufficient kinetic energy to rupture the insulating native oxide layer on impact, while particles above that range will rupture, merge, and form a conductive film. The methods characterized here have implications for facile patterning of highly elastic metallic conductors for soft and stretchable electronics, e-skins, and soft robotics.

## 4. Experimental

### 4.1. Materials

The liquid metal used here is a eutectic alloy of gallium and indium (75.5 wt % Ga, 24.5 wt % In), purchased from The Indium Corporation. The liquid metal has a melting point of 15.5 °C [68,90]. Thus, it is liquid at room temperature. The metal forms a thin native oxide, which provides a mechanical shell around a fluid that otherwise has a very high surface tension (~640 mN·m^−1^) [68] and low viscosity (~2 mPa·s) [91].

Silicone elastomers used in this work are Sylgard 184 and Sylgard 186 (Dow). All silicones were initially prepared at a 10:1 ratio of base:curing agent before dilution with toluene to the desired concentration.

Liquid metal particle suspensions were formed via sonication (for nanoparticles) or mechanical mixing (for microparticles). For sonication, 1 g of bulk EGaIn was placed in a 20 mL vial with 15 mL of ethanol. This solution was sonicated using a Q700 Probe Sonicator (QSonica Instruments, Newtown, CT, USA) for 10 min at 50 amps and 10 amps to produce nanoparticles (1 and 2 µm in diameter respectively). Microparticles were prepared by placing the same EGaIn/ethanol solution on an Analog Vortex Mixer (VWR) for mechanical mixing. The largest particles (191 µm) were mixed at full power for 1 min, and the next set of particles (15 µm) were mixed for 5 min.

### 4.2. Stencil Preparation

Vinyl stencils (ORACAL 651) were prepared using a Cameo 3 Desktop Cutting System (Silhouette America, Lindon, UT, USA). Depending on the complexity of the pattern the stencils were adhered directly to the substrate or applied using transfer tape.

### 4.3. Aerosol Deposition

EGaIn was deposited in these experiments by aerosol jetting through an airbrush nozzle with compressed air. Air was taken from a compressed air line at 30 psi. The airbrush used was a Model G22 Gravity Feed Airbrush (Master Airbrush).

### 4.4. Surface Profiling

Surface profiles of liquid metal and polymer films were determined via white light interferometry using a Profilm3D optical profiler (Filmetrics, San Diego, CA, USA) using a 10× objective lens.

### 4.5. Electrical Measurements

Electrical measurements were conducted using a Keithley 2400 Sourcemeter (Keithley Instruments, Cleveland, OH, USA). For the strain sensor measurements, a constant voltage of 1 V was applied while the samples were strained. The capacitance of the capacitive sensor was measured using a handheld LCR meter (Agilent U1733C, Agilent Technologies, Inc., Santa Clara, CA, USA).

### 4.6. Viscosity Measurements

The viscosity of the silicone solutions was measured using an AR-G2 rheometer (TA Instruments). The viscosity of pure PDMS (Sylgard 184 or Sylgard 186) were measured using a parallel plate geometry with a 40 mm diameter. The solutions of PDMS in toluene were measured with a concentric cylinder geometry. The reported data is the high-shear viscosity, gathered at a shear rate of 100 s^−1^.

### 4.7. Force Measurements

Force measurements were conducted using an Instron 5493 Universal Testing System (Instron) in compression mode. For measuring the critical force to induce conductivity in EGaIn particles, a modified probe with two copper electrodes attached to the face was used. The applied force was measured using a load cell rated for up to 1 kN of force.

The PDMS samples for force measurements were prepared by first spin-coating a sacrificial layer of polyvinyl alcohol (PVA) on a glass slide. A solution of 20 wt % PVA in water was spun at 500 rpm for 20 s, followed by 1000 rpm for 10 s to produce a uniform layer. Then, we sprayed PDMS onto the glass slides, followed by curing at 70 °C for at least 48 h to ensure the films were completely cured and no solvent remained within the film. Finally, the sample was placed in water to dissolve the PVA layer, freeing the PDMS layer from the slide.

## Figures and Tables

**Figure 1 micromachines-12-00146-f001:**
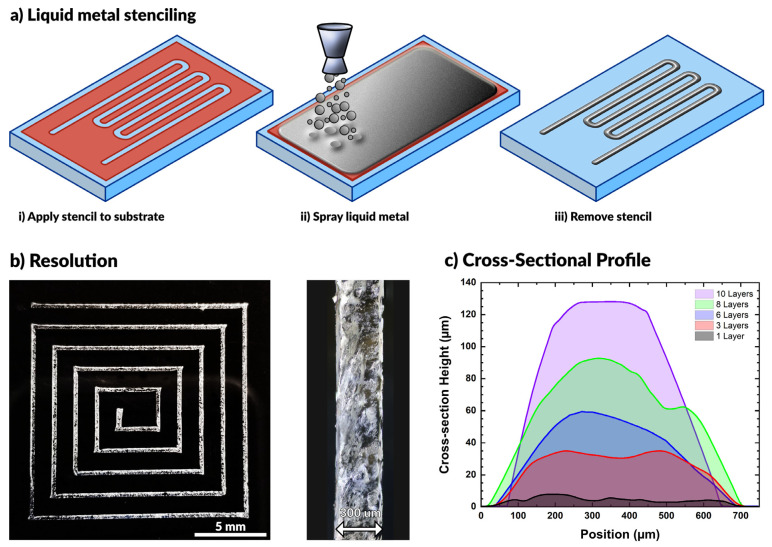
(**a**) Schematic depiction of the aerosol spraying process for liquid metals. Vinyl stencils conform to the substrate, the airbrush sprays liquid metal across the surface, and removal of the stencil produces a pattern. (**b**) A square spiral patterned by spraying liquid metal with 300 μm wide lines. (**c**) Cross-sectional profile of liquid metal lines after multiple coatings.

**Figure 2 micromachines-12-00146-f002:**
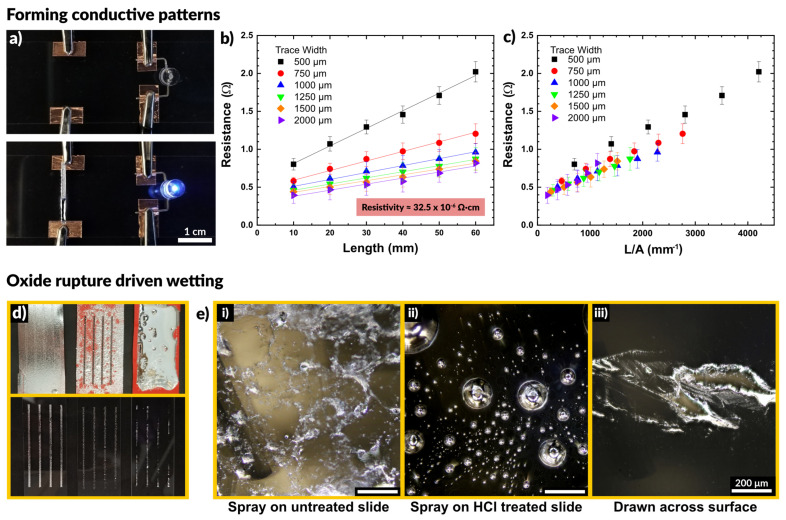
(**a**) A sprayed line of metal connects a circuit to power an LED, demonstrating that sprayed traces are immediately conductive without post-processing. (**b**,**c**) Plots of resistance as a function of trace length and trace width. Resistivity is calculated by determining a line of best fit and using Pouillet’s law. (**d**,**e**) A comparison of depositing liquid metal through stencils by spraying on an untreated glass slide, spraying on an HCl treated slide, and drawing liquid metal across the surface. The spray coated sample is uniformly covered, while the HCl treated slide show disconnected droplets and the drawn sample shows non-uniform wetting over the surface.

**Figure 3 micromachines-12-00146-f003:**
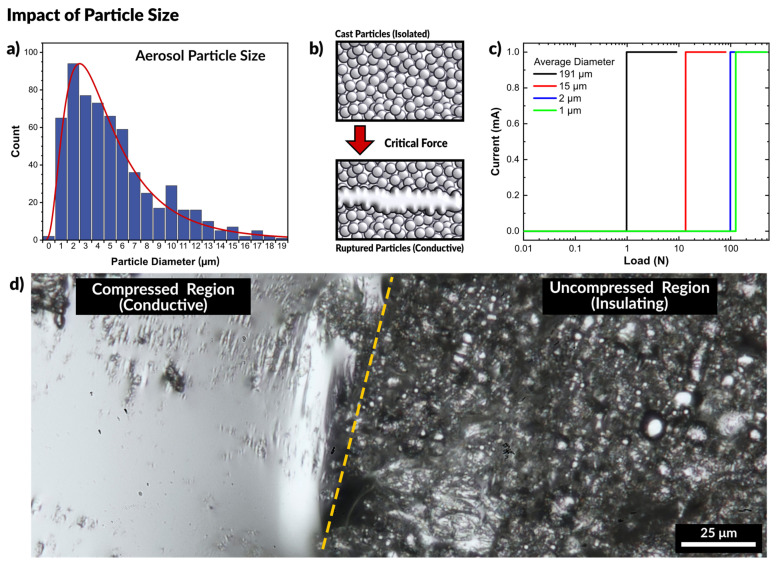
(**a**) Particle size distribution of aerosolized liquid metal. (**b**) Schematic of experiment where force is applied to a cast film of liquid metal particles. With sufficient force the particles rupture and merge to form a conductive path. (**c**) Plot of measured current across a liquid metal particle film as a function of applied pressure. Smaller particles require greater force to rupture and form a conductive path, seen by the load at which the current increases by a step-change. (**d**) Optical micrograph of the boundary between the compressed region and uncompressed particles of a cast film. The compressed region is conductive while the uncompressed regions remain insulating.

**Figure 4 micromachines-12-00146-f004:**
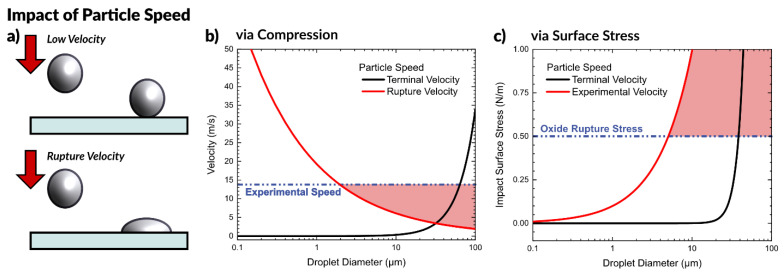
Impact of particle speed at impact. (**a**) Depiction of a droplet hitting a surface at low velocity and coming to rest, compared to a droplet hitting a surface at a velocity sufficient to rupture the oxide, deform the drop, and pin it to the surface. (**b**) Graph to predict the required rupture velocity as a function of particle size, compared with the terminal velocity of a particle and the experimental velocity of aerosolized liquid metal particles. (**c**) Graph of predicted surface stress for a particle at impact as a function of particle size at experimental velocity and terminal velocity. In both plots, regions highlighted in red are conditions that create particles with sufficient kinetic energy to rupture on impact.

**Figure 5 micromachines-12-00146-f005:**
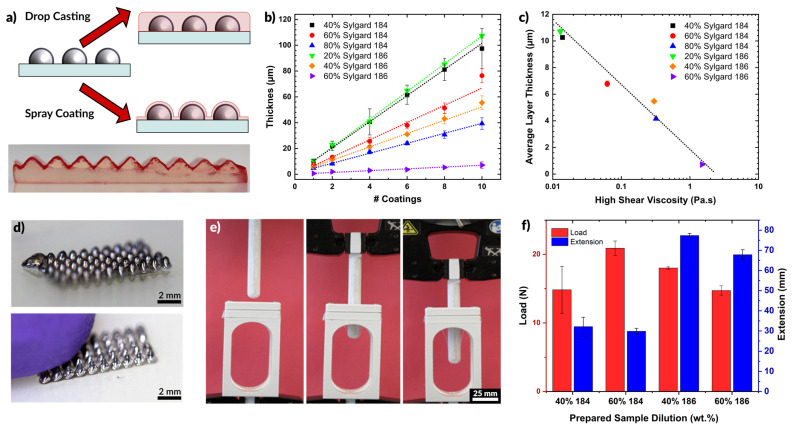
(**a**) Schematic depiction of coating of silicone. Drop casting produces a uniformly level surface while spray coating follows the profile of existing features. PDMS colored with red dye is shown coating an un-dyed PDMS substrate with topography. (**b**) Layer thickness of cured PDMS films for varying dilutions of Sylgard 184 and Sylgard 186. The layer thickness increases with additional coatings. (**c**) Plot of the average layer thickness for each sample in (**b**) plotted against the viscosity of the dilute solution (i.e., before spraying). Layer thickness decreases for higher viscosity fluids. (**d**) Liquid metal cones, deposited by printing, which are encased in PDMS by spray coating. The encapsulation is optically transparent, but robust to protect the liquid metal from the probing finger. (**e**) PDMS film strength experimental setup. A film is loaded between two clamps at the top of the open structure and the probe is lowered down measuring the force and extension. (**f**) Bar plot showing the average load and extension at rupture for a set of 200 μm PDMS films prepared by spray coating.

**Figure 6 micromachines-12-00146-f006:**
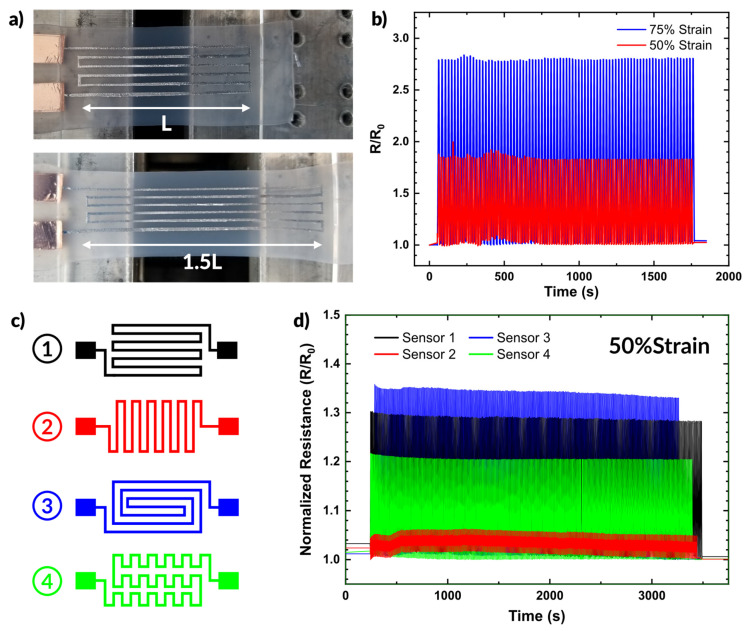
Applications of aerosol patterned devices. (**a**) Photographs of a sprayed liquid metal sensor being strained. (**b**) Plot of resistance over time during strain cycling at 50% and 75% strain. (**c**,**d**) A set of strain sensors with different geometries, and the impact the change in geometry has on the resistance during strain cycling plotted over time.

**Figure 7 micromachines-12-00146-f007:**
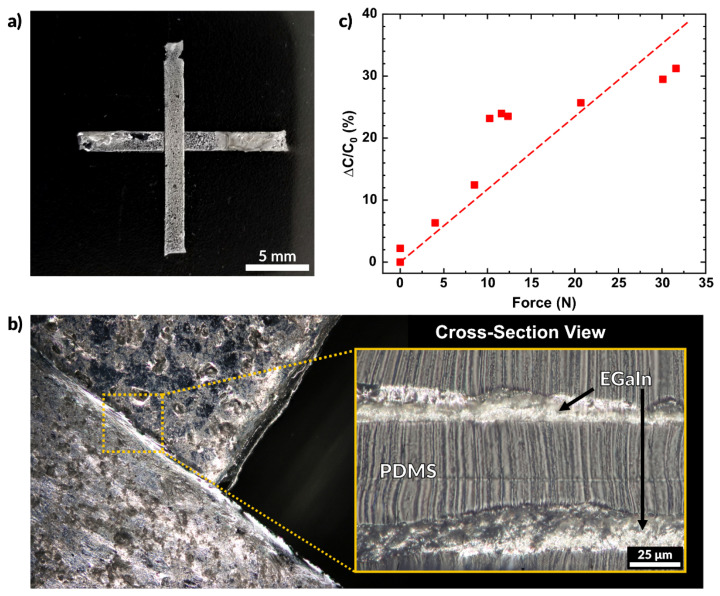
A capacitive sensor created entirely by aerosol spray coating. (**a**) Multilayer capacitive sensor made using liquid metal electrodes and sprayed layers of PDMS. (**b**) Overhead view of the overlap of the two electrodes and a cross-sectional view of the electrodes separated by a layer of sprayed PDMS. (**c**) Plot of the measured change in capacitance as a function of force.

## Data Availability

Data available on request from the authors.

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
