# Peer review of "Aerosol Spray Deposition of Liquid Metal and Elastomer Coatings for Rapid Processing of Stretchable Electronics"

_micromachines, 2021, doi:10.3390/mi12020146_

Round 1

Reviewer 1 Report

The authors present a detailed investigation into spray coating of liquid metal with stencil patterning. The coating process and underlying droplet dynamics are investigated in detail and example applications such as strain sensors and capacitive force sensors highlight utility of the technique.

The manuscript is timely, well written and experiments support all claims and analysis in the manuscript. I recommend the publication with minor changes. The sensor section is a little light in comparison to the rest of the manuscript. It would be great if a little more detail on the sensor characterization could be added.

Specific comments:

  • Excellent intro with broad overview of the field and motivation for this work.
  • Great characterization of the spray coating process
  • Figures look very pixelated and are difficult to read as a result, I suspect it is the pdf. conversion of the online system, high res versions should be used for the published article.
  • The strain sensor section is a little light on the discussion. There are a couple of features of the strain response that are not discussed and should be addressed. This includes:
    • Why is there a baseline drift for some of the sensor designs?
    • Where do the spikes in signal come from?
    • Given this relatively unstable response to strain, what is the cyclic stability of these sensors, it would be great if a test with 100k cycles is shown.
  • Same applies for the capacitive sensor characterization, the response seems to be very nonlinear and its not clear how the test was done (what was the object shape that applied the force, where exactly was the force applied etc.)
    • Here it would be great to show repeatability of the results.
    • What is the variation between devices? Maybe 3 devices could be compared.

Reviewer 2 Report

Please look at the attachment. 
